# Partial Polymer Blend for Fused Filament Fabrication with High Thermal Stability

**DOI:** 10.3390/polym13193353

**Published:** 2021-09-30

**Authors:** Muhammad Harris, Johan Potgieter, Hammad Mohsin, Jim Qun Chen, Sudip Ray, Khalid Mahmood Arif

**Affiliations:** 1Massey Agrifood Digital Lab, Massey University, Palmerston North 4410, New Zealand; j.potgieter@massey.ac.nz; 2Industrial and Manufacturing Engineering Department, Rachna College of Engineering and Technology, Gujranwala 52250, Pakistan; 3Department of Polymer Engineering, National Textile University, Faisalabad 37610, Pakistan; mhammad@ntu.edu.pk; 4School of Food and Advanced Technology, Massey University, Palmerston North 4442, New Zealand; q.chen2@massey.ac.nz; 5New Zealand Institute for Minerals to Materials Research, Greymouth 7805, New Zealand; sudip.ray@nzimmr.co.nz; 6Department of Mechanical and Electrical Engineering, SF&AT, Massey University, Auckland 0632, New Zealand; k.arif@massey.ac.nz

**Keywords:** fused deposition modeling, polypropylene, polylactic acid, thermal aging, degradation, pellet printing

## Abstract

The materials for large scale fused filament fabrication (FFF) are not yet designed to resist thermal degradation. This research presents a novel polymer blend of polylactic acid with polypropylene for FFF, purposefully designed with minimum feasible chemical grafting and overwhelming physical interlocking to sustain thermal degradation. Multi-level general full factorial ANOVA is performed for the analysis of thermal effects. The statistical results are further investigated and validated using different thermo-chemical and visual techniques. For example, Fourier transform infrared spectroscopy (FTIR) analyzes the effects of blending and degradation on intermolecular interactions. Differential scanning calorimetry (DSC) investigates the nature of blending (grafting or interlocking) and effects of degradation on thermal properties. Thermogravimetric analysis (TGA) validates the extent of chemical grafting and physical interlocking detected in FTIR and DSC. Scanning electron microscopy (SEM) is used to analyze the morphology and phase separation. The novel approach of overwhelmed physical interlocking and minimum chemical grafting for manufacturing 3D printing blends results in high structural stability (mechanical and intermolecular) against thermal degradation as compared to neat PLA.

## 1. Introduction

Fused filament fabrication (FFF) or fused deposition modeling (FDM) is a renowned additive manufacturing (AM) technique [1,2]. FFF is common among the commercial and domestic users due to its various benefits like low cost, good resolution, high mechanical properties, etc. [3,4,5]. In recent developments, the FFF is considered as one of the potential techniques in AM for large-scale applications [6,7,8]. In this regard, the pioneering work is reported as big area additive manufacturing (BAAP) that utilizes acrylonitrile butadiene styrene (ABS) with carbon fibers [9]. The benefits gained from addition of carbon fibers include the mechanical strength and 3D printing with least polymer warpage [9]. On the contrary, there are numerous other structural requirements that are not yet considered properly in the literature for large-scale AM. For example, the mechanical stability of any FFF material in terms of strength has not yet been evaluated in severe environments like high temperature. 

Polylactic acid (PLA), a biodegradable polyester, is recognized as a commodity polymer in FFF [10]. The reason for using PLA over other common FFF materials has been widely accepted through extensive research that unveils numerous reasons, e.g., it is eco-friendly, has high mechanical properties, and has good printability [3,10]. Particular to the mechanical strength, the tensile strength for PLA (55 MPa) is high as compared to ≈41 MPa of ABS, ≈45 MPa for polycarbonate (PC), ≈38 MPa for ABS/PC, ≈35 MPa for Nylon, ≈38 MPa for polypropylene (PP) [3]. However, as being a polyester, the backbone of PLA main chain is vulnerable to chemical scission due to poor thermal stability [11,12]. The recent work from the authors of this research has found visible degradation of C-O-C and C=O groups from Fourier transform infrared spectroscopy (FTIR) analysis due to thermal aging [12]. This kind of structural deficiency can cause detrimental effects in large-scale AM applications aimed for real environments. This highlights the need to explore the ways to make PLA stable for large-scale FFF applications while maintaining its biocompatibility. 

One of the common approaches to achieve structural stability is “melt blending” of PLA with high temperature non-biodegradable polymers (nylon, ABS) in the presence of different additives (thermal stabilizers, fillers, fibers) [13,14]. The researched blends of PLA with non-biodegradable polymers provide reasonable mechanical properties; however, the high composition (>25%) of non-biodegradable polymers to achieve optimal properties [12,13,14] is still a big challenge to make the blend eco-friendly as neat PLA. In this regard, one of the recent blends of PLA with less than 10% high density polyethylene (HDPE) presents good thermal properties as compared to neat PLA [12]. 

To achieve good thermal properties in PLA, the literature reports melt blending of PLA with PP compatibilized with PP-g-MAH (polypropylene graft maleic anhydride) in the presence of natural fibers (hemp, Harakeke). However, the non-biodegradable component (PP) is in high percentage (52.5%), with just 22.5% of biodegradable component (PLA). Moreover, the tensile strength is limited to meagre 28.1 MPa for a fiber composite [15]. The high composition (52.5%) of fossil fuel-based polymer (PP) makes the final blend non-biodegradable, and low AM strength [15] makes the composite unsuitable for real life applications like large-scale additive manufacturing. It is also noted that the compatibilization of PP and PLA with PP-g-MAH is not able to achieve requisite high mechanical properties even at room temperature (28 MPa) [15]. Furthermore, the properties will probably further degrade in the vulnerable environments (thermal aging) due to the weak PLA intermolecular structure. Therefore, this highlights the novel research proposition for finding a suitable processing route to make a PLA/PP blend stabile without use of reinforced fibers. 

This research aims to explore the intermolecular approach of partial grafting and physical interlocking for PLA and PP blend in FFF that is partially compatibilized by HDPE-g-MA. Based on the principles of blending [13], PP in the presence of HDPE-g-MA will show further increases in physical interlocking due to the incompatibility between PP and HDPE, which has not yet been explored for the effects on the overall stability of FFF materials. This research also presents full factorial statistical analysis for thermal degradation of the blend. The statistical design of experiments also analyzes the effects of combined in-process temperatures (bed temperature, printing temperature) and thermal degradation mechanisms. 

## 2. Materials and Methods

### 2.1. Materials

Extrusion grade 2002D of PLA is provided by Scion, Rotorua, New Zealand with a specific weight of 1.24 g/cm^3^. HDPE-g-MAH is purchased from Shenzhen Jindaquan Technology Co. Ltd, Shenzhen, China. The composition of HDPE-g-MAH is 95:5 by weight. Moplen HP400N polypropylene homopolymer with a specific weight of 0.905 g/cm^3^ and melt flow index (MFI) of 11 g/10 min is procured from TCL Hunt Auckland, New Zealand. Based on the objective of achieving excessive physical interlocking and limited chemical grafting between three polymers (PLA, PP and HDPE-g-MAH), a high melt flow index grade of PP is selected to ensure better melt blending [13]. The high MFI also helps to achieve probable chemical grafting easily as compared to low MFI [10,13].

### 2.2. Melt Blending

The materials are blended in a single screw extruder. The criterion for making different compositions is based on the ability of each composition’s 3D printing. Each blend composition is printed after extrusion before moving to the fabricating of next composition. The compositions of the non-biodegradable polymer (PP) and compatibilizer (PE-g-MAH) in PLA, for the first blend, are based on minimum corresponding weight compositions in the literature [16,17,18,19,20]. In this regard, the first extrusion is performed with 20% PP [16,17] and 5% PE-g-MAH [18,19,20]. Large die swelling is obtained during 3D printing with 2.35 mm of bead instead of the requisite 0.2 mm. The die swelling is probably due to the negative effects of large composition for MAH [12]. 

The second blend composition is decided with minimum percentage composition of non-biodegradable polymer (PP) and PE-g-MAH to control the die swelling. The minimum percentage is decided to solve issues like die swelling and printability. Furthermore, one of the main objectives is to make the blend eco-friendly. In this regard, the minimum percentages of 7.5% non-biodegradable polymer and 0.5% compatibilizer [12,20] are selected. The second composition is 3D printed with negligible die swelling and warpage. Therefore, further blend compositions are not fabricated. The compositions are given in Table 1.

All three polymers (PLA, PP, PE-g-MA) are dried in a thermostat blast oven (HST, Shanghai, China) for 1 h at 40 °C followed by mixing in a mixer for five minutes. Polymer compounding of three polymers was performed in a single screw extruder (HAAKE^TM^ Rheomex OS by Thermo scientific) at Scion, New Zealand. A single screw extruder was used to achieve minimum degradation of the polymer blend before 3D printing [21]. Avoiding the degradation in the single screw extruder will ensure the true analysis of degradation during 3D printing followed by thermal aging. The temperature of zones from feeder to extruder nozzle are: 170 °C, 175 °C, 185 °C, 185 °C, 185 °C, 185 °C, 180 °C, 180 °C, 175 °C, and 145 °C. The extrusion is performed at 100 rpm and 70−90 bar of a die pressure. The pelletizer was operated at 13.5 mm/min to achieve the approximately 1.3 mm long pellets. 

### 2.3. Pellet 3D Printing

3D printing is performed at our in-house built pellet printer [22] customized for PLA/PP/PE-g-MAH. Generally, the polymer extruders consist of three zones: (1) Feeding, (2) compression (mixing), and (3) metering [22]. However, the screw configuration in the pellet printer is particularly designed with only feed zone that avoids thermo-mechanical shear as shown in Figure 1. Additional shear is reported with thermo-mechanical degradation of polymers in literature [22,23,24]. The thermal degradation is also managed with the help of a liquid cooling chamber and teflon insulated partition assembled above the heating barrel presented in Figure 1. Furthermore, the aluminium hopper is fitted with an SLS printed cone. The polymeric SLS cone also acts as a heat separator to avoid the heat propagation from hot barrel to the raw pellets. 

The combined assembly of the teflon insulator, cooling jacket, and SLS separator avoids the excessive heating of raw pellets in the upper part of feeder before extrusion. Moving another step ahead to avoid thermo-mechanical shear, the screw used for this research has variable length. The length in this case is set at ≈15 mm away from the base of the heating barrel, as shown in Figure 1. The larger distance between tip of the screw and base manages more melt pool that is extruded with least possible thermo-mechanical shearing from a chamfered barrel base under the action of gravity (Figure 1). 

An opensource slicing software (Slic3r) is used to slice the “stl” (Standard Tessellation Language) CAD files of ASTM D638 type IV dogbones [25]. Other parameters include: Layer thickness 0.2 mm, multiplier 5, nozzle diameter 0.4 mm [11], infill density 100% [3], and infill pattern 45°/−45° [3]. The printer is operated using “Pronterface” which uses the G-codes files from the “Slic3r”.

### 2.4. Thermal Analysis

The thermal analysis is designed for evaluating the effects of two types of thermal treatments: (1) low to high temperatures during 3D printing [12], and (2) post-printing aging [26]. Two variables (bed and printing temperatures) [12,27] are selected for thermal analysis of temperature during printing. The bed temperatures of 25 °C, 55 °C, and 85 °C are used. The bed temperature of 25 ± 2 °C is selected based on the lowest possible temperature at ambient environment. The highest bed temperature of 85 ± 2 °C is selected for being higher than the glass transition temperature of PLA (≈55) [28] that can potentially affect the intermolecular structure [12,29]. The printing temperature is selected based on minimum and maximum extremes obtained during preliminary 3D printing trials. As it is shown in Figure 2a, the clogging is observed near a temperature of 155 °C, causing no extrusion of the polymer blend at all. On the high end, the temperature of about 177 °C results in severe degradation of polymer beads and interrupted melt flow, as shown in Figure 2b. Therefore, the printable temperature range of 161 °C to 171 °C is selected for pellet 3D printing.

The post-printing aging temperature of 75 ± 3 °C is set for 15 days to thermally age the ASTM dog bones. The temperature of 75 ± 3 °C is selected based on the reported degradation of PLA for more than one hour of aging above glass transition temperature (≈55–65 °C) [12,16,29,30].

All aforementioned variables (bed temperature, printing temperature, and thermal aging) are designed with “general full factorial ANOVA with multiple levels” to analyze the significance of in-process and post-printing temperature (aging) treatments. The factors and levels for thermal ANOVA are provided in Table 2. 

### 2.5. Tensile Testing

All samples are tested for tensile strength and strain on Instron 5967 at an extension rate of 5 mm/min. The quasistatic loading (tensile loading) is applied using a load cell of 30 kN with a clip-on-gauge extensometer of 25 mm. The average of multiple samples is taken for tensile strength and strain in ANOVA analysis. 

### 2.6. Fourier Transform Infrared Spectroscopy (FTIR)

Fourier transform infrared spectroscopy (FTIR) is used to analyze the effects of thermal aging on intermolecular chains and bonds. In this regard, a Thermo electron Nicolet 8700 FTIR spectrometer is used to scan and collect the FTIR spectrum of the fractured samples in the range of 400−4000 cm^−1^. The total of 30 scans are averaged by the OMNIC E.S.P software version 7.1 to generate a single FTIR spectrum for each sample. Furthermore, OMNIC E.S.P is also used to perform the normalization and correction with respect to the base line for each spectrum. 

### 2.7. Differential Scanning Calorimetry (DSC)

Differential scanning calorimetry (DSC) is used to analyze the nature of blending (grafting or physical interlocking) and the expected effects of thermal treatments (aging). The analysis is based upon corresponding temperatures and enthalpies of glass transition, cold crystallization, melt crystallization and degradation. NETZSCH simultaneous thermal analyzer (STA) 449 F1 Jupiter (Germany) is used with a nitrogen purging at a flow rate of 50 mL/min. The range of thermal analysis is from 25 °C to 550 °C with a rate of 10 °C/min.

### 2.8. Thermogravimetric Analysis (TGA)

Thermogravimetric analysis (TGA) is used to further analyze and validate the nature of melt blending (grafting or physical interlocking). STA 449 F1 Jupiter from NETZSCH, Germany is operated in a range of 25 °C to 550 °C at a nitrogen purge flow rate of 50 mL/min. The rate of temperature increase is 10 °C/min. 

### 2.9. Scanning Electron Microscope (SEM)

A scanning electron microscope (SEM) is used to perform a visual analysis of any form of degradation or nature of blending (phase separation). The fractographic analysis for fractured samples is performed on a Hitachi TM3030 Plus desktop SEM (Japan). All samples are analyzed at the fractured cross section at variable magnifications.

## 3. Results

The ANOVA analysis for thermal effects on stability (tensile strength) is shown in Figure 3. All potential thermal degradation variables (bed temperature, printing temperature, and thermal treatment) were found to be insignificant. The nearest single variable to 0.05 (5%) confidence level was thermal aging treatment with *p*-value of 0.073. The significance is shown for binary interaction of printing temperature and thermal aging (Figure 3a) with a *p*-value of 0.037. 

The “main effects plot” (Figure 3b) for thermal analysis reveals the decrease of tensile strength for thermally treated (aged) samples as compared to the non-treated samples. This shows that the thermal degradation is visible, however, it is not significant as obtained in the pareto chart (Figure 3a). The lowest obtained for PLA/PP/PE-g-MAH blend (37 MPa) after 15 days of aging is still higher than the degraded neat PLA (34 MPa) [12]. This shows the stability of the novel blend against thermal aging.

## 4. Discussion

### 4.1. Analysis for Thermochemical Effects Using FTIR

The intermolecular interactions are analyzed using FTIR as shown in Figure 4. The graphs obtained for neat PLA, PP, and PE-g-MAH are similar as reported in corresponding literature. PLA is confirmed with the three chemical groups of C-O-C, C=O and C-H [31,32,33] obtained at 1085 cm^−1^, 1747 cm^−1^ and 2995−849 cm^−1^ respectively. The stretching vibrations (asymmetric and symmetric) for CH_2_ and CH_3_ [34,35,36] is noted for PP in the range of 2800–3000 cm^−1^. The graft co-polymer of PE-g-MAH is confirmed with distinct peaks for: (1) saturated hydrocarbon (C-H) of high density polyethylene at 2915 cm^−1^ and 2848 cm^−1^, (2) MAH by C=O group at 1705 cm^−1^ and 718 cm^−1^ [12,32]. 

The effects of blending and 3D printing of non-treated combinations on intermolecular interactions at low (161 °C, 25 °C) temperature are compared with the neat PLA, PP and PE-g-MAH in Figure 4. The effects of melt blending are observed in three forms: (1) chemical shifting of various groups, (2) variation in transmittance percentage (intensity), and (3) formation of a distinct peak. The shift of following chemical groups shows the intermolecular interactions in the non-treated blend: saturated hydrocarbon (C-H) groups [12,31] in neat PLA at 2996 cm^−1^ to 2988 cm^−1^, C-O-C groups [12] in neat PLA at 1085 cm^−1^ to 1082.2 cm^−1^, and C=O groups [12] in neat PLA at 1747 cm^−1^ to 1741 cm^−1^. Furthermore, the variations in the intensity of transmittance show intermolecular interactions. For example, the intensity of C=O [12] at 1746 cm^−1^ in the PLA/PP/PE-g-MAH is increased after single screw extrusion as compared to the neat PLA. This is probably due to the synchronized effects of C=O groups of MAH at 1705 cm^−1^ as noted with a prominent hump magnified in a rectangle (Figure 4). The third form of intermolecular interactions after blending is found with the appearance of distinct saturated hydrocarbon (C-H) [37] peak of PP merged with the C-H groups of PLA. The three transmittance peaks related to C-H groups of PLA in non-treated blend at (161 °C, 25 °C) appeared with a distinct fourth peak around 2950 cm^−1^ (Figure 4). Furthermore, all four C-H peaks in non-treated blend present drastic decrease of transmittance percentage (Figure 4). The fourth distinct peak depicts the phase separation of PP in PLA with decreased transmittance (≈88% in PP to 98% in blend). The decrease in transmittance shows highly restricted intermolecular mobility [38]. 

The following discussion will describe the effects of thermal treatment (aging) at the lowest (161 °C, 25 °C) and highest (171 °C, 85 °C) temperature combination (Figure 5). The effects of thermal treatment (aging) on intermolecular interactions are analyzed via comparison of treated (aged) combinations at low and high temperatures with non-treated combinations at the corresponding temperatures (low and high) in Figure 5. The low temperature treated combination (161 °C, 25 °C) shows the reduction of −1% (85–86%)) in C-O-C groups and −5% (85–90%) in C=O groups. The minor decrease of C-O-C groups, showing chain scission [39], and the high depletion of C=O groups, showing the probable chemical grafting with MAH [12], thus provide a chemical justification for high strength at low printing temperature (161 °C) in main effects plots (Figure 3b). On the contrary, the high temperature combination (171 °C, 85 °C) presents −1.7% (88–89.7%) of C-O-C and −0.8% (91.2–92%) of C=O. The reduction of 1.9% as compared to 1% in C-O-C and 0.8% as compared to 5% in C=O shows comparatively more chain scission [39] and less chemical grafting [12], respectively. This explains the decrease of tensile strength in “main effect plots” after thermal treatment (Figure 3). 

The chemical interactions are prominently apparent in FTIR analysis. However, the minor changes of 1% C-O-C groups in low temperature combinations and 0.8% of C=O in high temperature combinations can be caused due to experimental error. Therefore, the thermally treated blend requires a separate thermal analysis of crystallization in different phases (glass, cold and melt) to understand the nature of chemical interactions (grafting or physical interlocking).

### 4.2. Analysis for Thermochemical Effects Using DSC

In DSC, the analysis for the effects of melt blending is provided in Figure 6 and Table 3. The chemical interaction between PLA and PP in the presence of a compatibilizer (PE-g-MAH) in reference blend pellets (non-printed) is observed in three phases, i.e., glass transition, melt crystallization and degradation. The glass transition temperature (T_G_) of the non-printed blend pellets is decreased to 63.2 °C from 65.5 °C of neat PLA. However, the enthalpy of glass transition (ΔH_G_) has increased significantly to 1.716 J/g as compared to 0.0261 j/g of neat PLA. The decrease in T_G_ shows the crystalline orientation at low temperature (63.2 °C) due to the physical interlocked PP [29,40] and the increase in ΔH_G_ shows the partial compatibilization [12,29]. Furthermore, the melt crystallization of non-printed blend pellets appears with distinct small steps on right hand side compared to the unimodal melt crystallization in neat PLA as shown in a magnified image (Figure 6). The melt crystallization temperature (T_M_) of blend pellets increases to 155.3 °C as compared to 152.4 °C of neat PLA (Table 3). The small distinct steps present immiscible [16,41] PP in PLA and the increased T_M_ point towards the probable partial (incomplete) compatibilization [12,42]. 

For in-depth analysis of the effects of printing temperature as noted in FTIR, the DSC analysis is also performed for non-printed blend pellets. The non-printed blend pellets are used as the “reference” for analyzing the effects of 3D printing at high and low temperatures. 

The degradation thermal profile of the blend pellets as compared to the neat PLA is observed with a decrease in ΔH_D_. This follows the general behaviour of immiscible blends [43]. Therefore, the prominent chemical interactions along with the immiscible PP shows the successful fabrication of a blend with partial compatibilization and physical interlocking. 

The thermal aging reveals extremely hard detection of glass transition and cold crystallization phases for low and high temperature combinations (Figure 6). This presents least to negligible chain orientation of PLA in PLA/PP/PE-g-MAH blend, which can be caused due to the excessive chain scission [12]. The FTIR analysis also verifies high chain scission of C-O-C groups (Figure 5).

### 4.3. Analysis for Partial Grafting and Physical Interlocking Using TGA

The thermal aging shows low T_ONSET_ of 338 °C at low temperature (161 °C, 25 °C) as compared to 350.3 °C of neat PLA. The high temperature (171 °C, 85 °C) presents almost similar T_ONSET_ of 348.4 °C as compared to the neat PLA (Figure 7). Similarly, the thermal aging at low temperature shows more decrease in temperature for mass loss from 50% to 80% as compared to high temperature combination (Table 4). The maximum decrease of “−2.18%” in temperature at low temperature combinations as compared to meagre “−0.56%” in high temperature combinations is a notable difference (Table 4). Despite thermal aging, the strange high onset temperatures and mass loss temperatures can be explained with the highest ΔH_D_ (864.3 J/g) among all combinations for blend that proves the synchronization of PP and PLA matrix. This shows the probable improvement in chemical grafting using the extra heat energy from high in-process temperatures (bed and printing) and aging [12,27]. The aging helps to react the non-grafted PP with PLA through PE-g-MAH that may remain unreacted during single screw extrusion. 

Thermogravimetric analysis (TGA) is performed to verify and quantify the nature of chemical interactions (grafting or physical interlocking) [12] as found in FTIR and DSC. TGA is also used to explain the stability of the novel blend against thermal degradation. 

The similar kind of post-printing grafting during thermal aging is reported for PLA and HDPE blend [12]. However, the contemporary blend is also found with 6.79% immiscible PP that presents an overwhelming physical interlocking above 400 °C as shown in Figure 8 and Table 4. Therefore, the reason for thermal stability as obtained in ANOVA analysis is the enhanced chemical grafting and physical interlocking.

### 4.4. Analysis for Visual Analysis of Melt Blending

Scanning electron microscopy (SEM) further validates the results found in the FTIR, DSC and TGA analysis with the visible physical interlocking. The overall morphology of the 3D printed blend at low temperature combination (161 °C, 25 °C) appears with poor adhesion between the layers as marked by circle in Figure 8a. This leads to low tensile strength for low temperature combination (161 °C, 25 °C). The increase of bed temperature (161 °C, 85 °C) shows uniform interlayer fusion but is still not sufficient as it appears in the form of a brittle fracture in Figure 8b. The morphology of low temperature combination (161 °C, 85 °C) presents “missing fractured beads”, which shows the poor adhesion due to lack of printing temperature [11]. The increase of printing temperature along with bed temperature (165 °C, 85 °C) improves the overall structural ductility as observed with long fibers [11] in Figure 8c. Furthermore, the physical interlocking of PP in PLA matrix is also observed in different forms. Figure 8d shows a phase separation in the pulled fibre. The non-reacted PP is enveloped in the PLA blend matrix as shown in Figure 8d. 

## 5. Conclusions

A chemical approach of a partially compatible blend with overwhelmed physical interlocking is presented for a novel blend of PLA/PP/PE-g-MAH for fused filament fabrication (FFF). The novel blend is prepared with the lowest PP percentage ever reported in the literature (7.5%), to enhance the stability of PLA against thermal treatment (aging). The blend is prepared in a single screw extruder to avoid any degradation due to thermal shearing. For 3D printing, the in-house built pellet printer is customized to print the material without any thermal shearing. A multi-level general full factorial ANOVA design of the experiment is designed for thermal degradation. Thermal degradation is analyzed for bed temperature, printing temperature and at 15 days thermal aging at 75 ± 3 °C aging. The following results are obtained:Based on the distinct C-H groups of PP in FTIR, immiscible PP in melt crystallization in DSC, and ≈6.2% to 6.75% immiscible PP in TGA, the PLA/PP/PE-g-MAH blend includes minor grafting and overwhelmed physical interlocking.Overall, the novel blend is stable against thermal degradation (aging) based on the insignificance found in ANOVA.The parameters found that significant thermal degradation is “printing temperature”. This confirms the thermal stability to thermal treatment (aging).The FTIR analysis presents the chain scission after thermal degradation. The chain scission occurs at the C-O-C bond. This chain scission is observed in DSC in the form of ΔH_M_ and ΔH_D_ for thermal degradation mechanism at any temperature combination. The chain scission is also supported with the decrease of onset temperatures in TGA analysis.

## Figures and Tables

**Figure 1 polymers-13-03353-f001:**
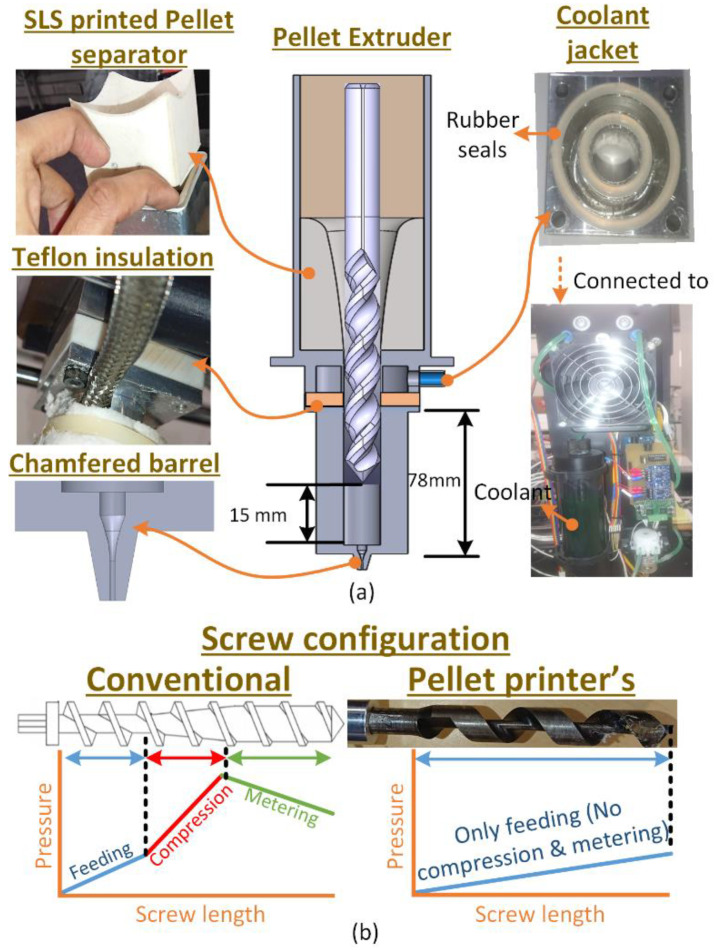
In-laboratory built pellet printer [22] customized for PLA/PP/PE-g-MAH blend; (**a**) types of customizations, and (**b**) difference between screw’s configuration of a conventional extruder (left) and customized pellet 3d printer (right).

**Figure 2 polymers-13-03353-f002:**
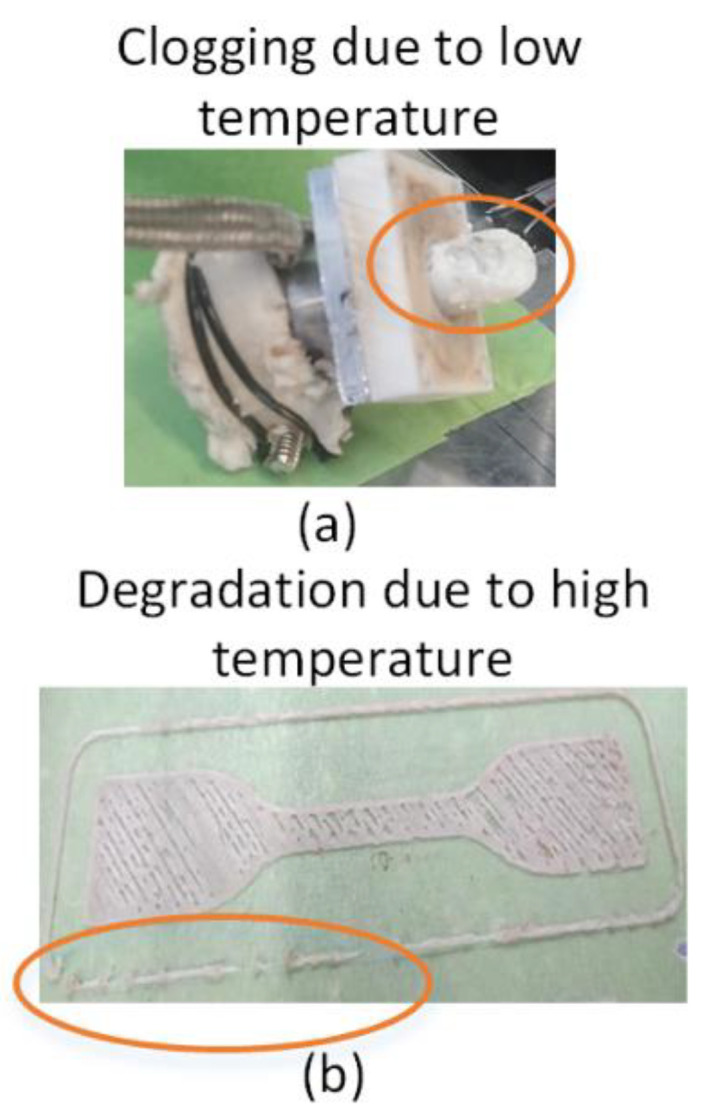
Selection of range for printing temperature: (**a**) 155 °C causes clogging, and (**b**) 177 °C causes thermal degradation.

**Figure 3 polymers-13-03353-f003:**
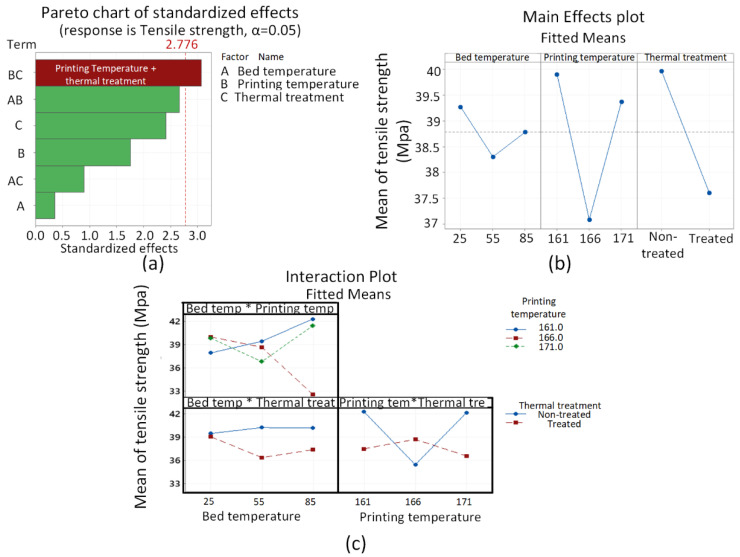
Results for thermal degradation: (**a**) Pareto chart, (**b**) main effect plots, and (**c**) interaction plots.

**Figure 4 polymers-13-03353-f004:**
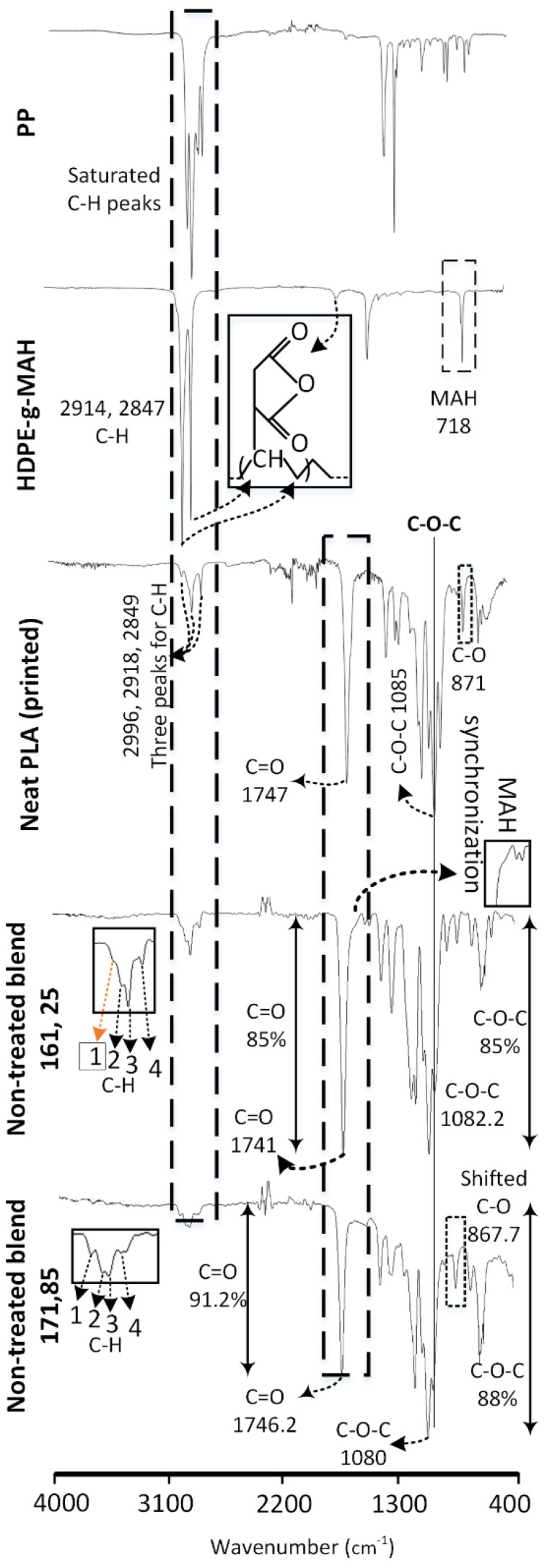
FTIR analysis for effects of melt blending on non-treated blend.

**Figure 5 polymers-13-03353-f005:**
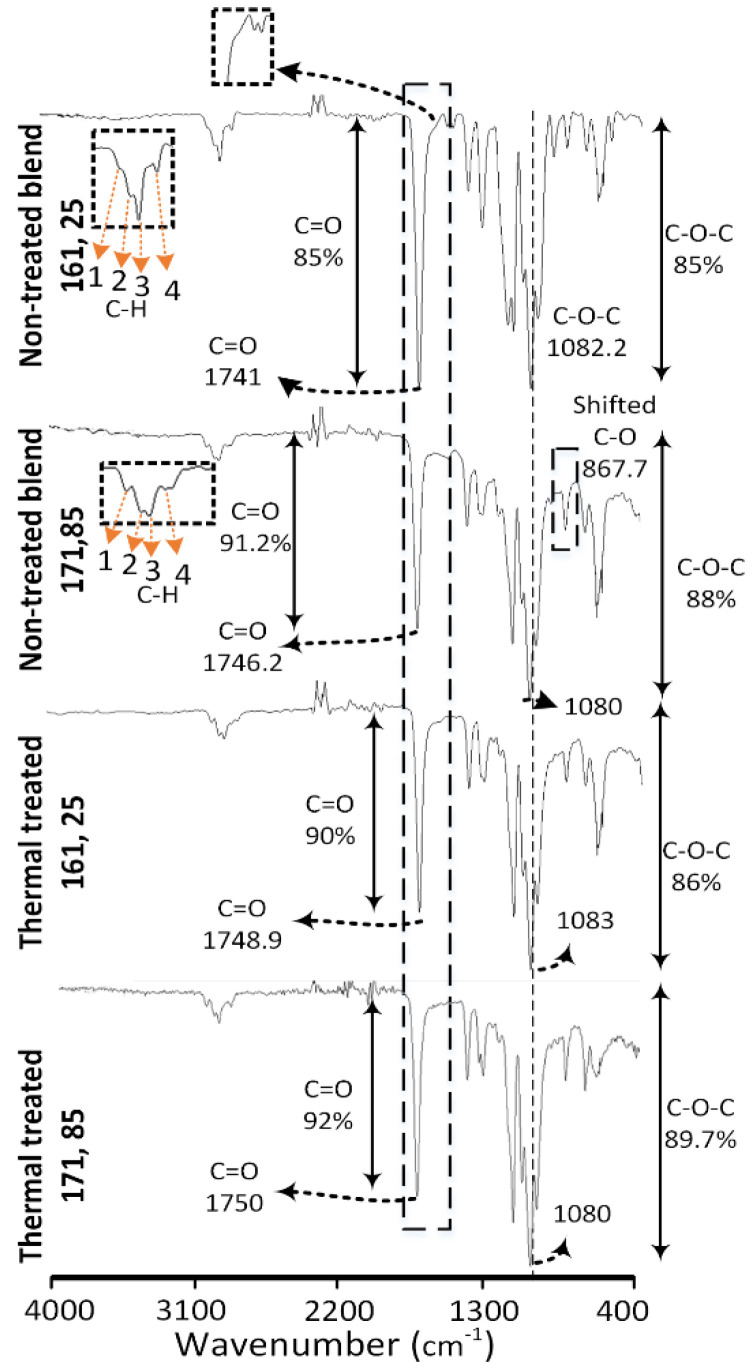
FTIR analysis for post 3d printing thermal treatment (thermal aging).

**Figure 6 polymers-13-03353-f006:**
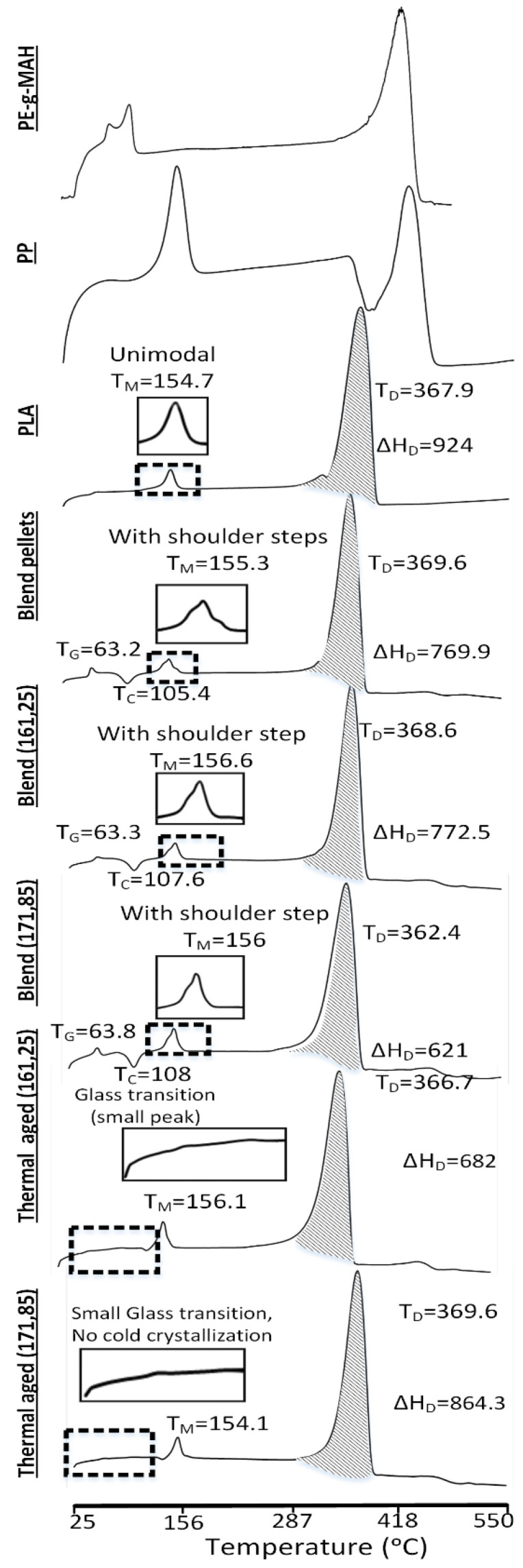
DSC analysis for effects of melt blending and thermal treatment (aging).

**Figure 7 polymers-13-03353-f007:**
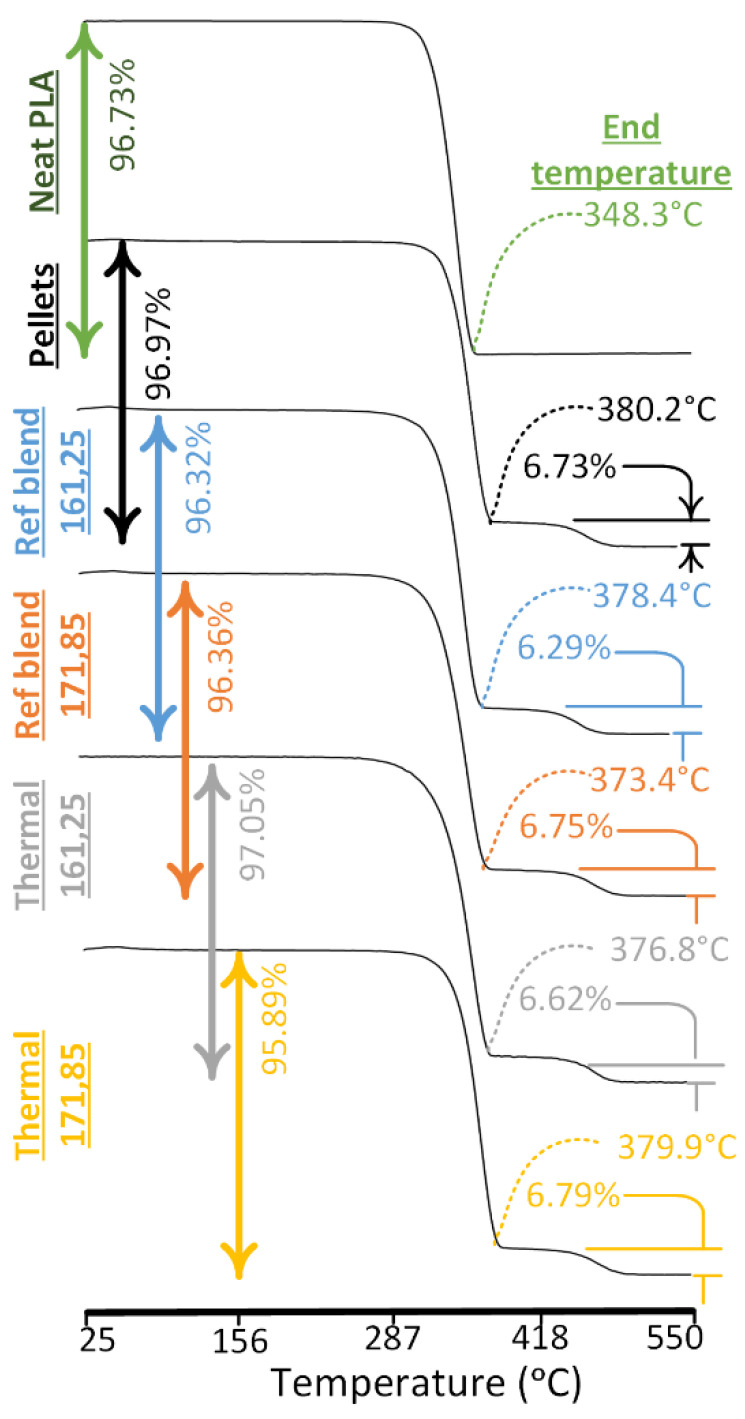
TGA analysis for effects of melt blending and thermal treatment (thermal aging).

**Figure 8 polymers-13-03353-f008:**
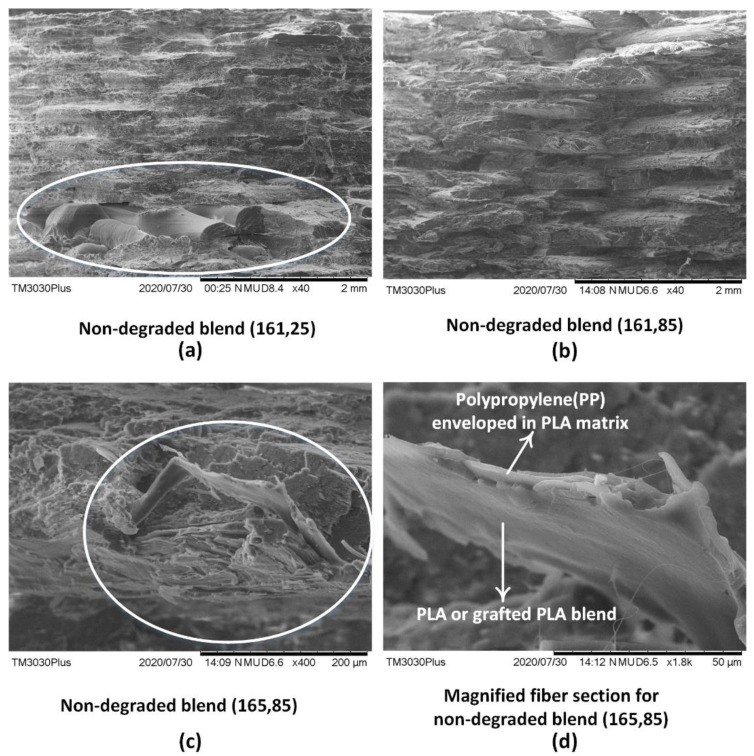
SEM analysis for: (**a**)non-degradable blend (161 °C, 25 °C), (**b**)non-degradable blend (161 °C, 85 °C) at 40X, (**c**) non-degradable blend at (161 °C, 85 °C) at 400X, and (**d**) magnified image of PP fiber (161 °C, 85 °C) at 1800X.

**Table 1 polymers-13-03353-t001:** Compositions prepared for ternary blend systems.

Blend	PLA	PP	HDPE-g-MAH
1	75	20	5
2	92	7.5	0.5

**Table 2 polymers-13-03353-t002:** Design of experiments for the analysis of thermal treatment (aging).

Analysis	Factor (Parameter)	Level 1	Level 2	Level 3
Thermal analysis	Bed temperature	25 ± 2 °C	55 ± 2 °C	85 ± 2 °C
	Printing temperature	161 ± 3 °C	166 ± 3 °C	171 ± 3 °C
	Thermal aging	0 days	15 days	

**Table 3 polymers-13-03353-t003:** DSC analysis for effects of blending, 3D printing, and thermal aging.

Materials	T_G_	ΔH_G_	T_C_	ΔH_C_	T_M_	ΔH_M_	T_D_	ΔH_D_
PP					170.6	82.5	458	148
PE-g-MAH					108.6	27.22	475.8	164.5
PLA(Non-treated)	65.5	0.0261	105.7	21.02	154.7	23.94	367.9	924
PLA/PE-g-MAH/PP pellets	63.2	1.716	105.4	11.73	155.3	11.96	369.6	769.9
PLA/PE-g-MAH/PP Reference (161,25)	63.3	1.645	107.6	13.17	156.6	12.7	368.6	789.6
PLA/PE-g-MAH/PP Reference (171,85)	63.8	3.086	108	11.52	156	13.64	362.4	621
PLA/PE-g-MAH/PP Thermal (161,25)	65.9	Negligible	105.9	Negligible	156.1	17.72	366.7	682
PLA/PE-g-MAH/PP Thermal (171,85)	66.6	Negligible	101.1 (not there)	Negligible	154.1	21.19	369.6	864.3

**Table 4 polymers-13-03353-t004:** Temperature for specific mass degradation. At each specific mass loss%, negative sign shows the decrease and positive sign shows the increase of corresponding temperature as compared to neat PLA.

Mass Loss%	PLA °C	Pellet, °C (% of PLA)	Non-Treated 161,25 °C (% of PLA)	Non-Treated 171,85 °C (% of PLA)	Thermal 61,25 °C (% of PLA)	Thermal 171,85 °C (% of PLA)
50%	367.7	365.9 (−0.49)	364.4 (−0.9)	356.1 (−3.15)	359.7 (−2.18)	365.8 (−0.52)
60%	371.2	369.3 (−0.51)	367.8 (−0.92)	360.4 (−2.91)	363.8 (−1.99)	369.2 (−0.54)
70%	374.7	372.8 (0.51)	371.2 (−0.93)	364.5 (−2.72)	367.9 (−1.81)	372.6 (−0.56)
80%	378.2	376.8 (−0.37)	375.2 (−0.79)	369.1 (−2.41)	372.2 (−1.59)	376.4 (−0.48)
90%	382.5	383.9 (−0.37)	383 (−0.05)	377.7 (−1.25)	378.9 (−0.94)	383.5 (+0.26)
92%	383.7	440.7 (+14.8)	443 (+15.4)	447.1 (+16.5)	440 (+14.67)	441.5 (+15.1)
95%	385.7	465.6 (+20.7)	465.9 (+20.8)	467.5 (+21.2)	465 (+20.6)	465.5 (+20.7)

## Data Availability

The data presented in this study are available in “Partial Polymer Blend for Fused Filament Fabrication with High Thermal Stability” at “https://www.mdpi.com/journal/polymers (accessed on 26 September 2021)”.

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
