# Peer review of "Partial Polymer Blend for Fused Filament Fabrication with High Thermal Stability"

_polymers, 2021, doi:10.3390/polym13193353_

Round 1

Reviewer 1 Report

I would recommend the paper “Partial polymer blend for fused filament fabrication with high  thermal stability” by Muhammad Harris, Johan Potgieter, Hammad Mohsin, Jim Quan Chen, Sudip Ray, Khalid Mahmood Arif, after minor revision. This paper seems to be a continuation of other works of these Authors.

Please take into account the remarks below:

  1. Would you please check the numbers of tables and figures in the text as they do not fit captions sometimes?
  2. What was the resolution during the recording infrared spectra as the data presented in Figs. 4 and 5 show decimal numbers.
  3. The 1% change in transmittance is within the experimental error, so maybe it is not worth comparing such minor changes.
  4. Page 9, lines 266-280. The Authors write about a reduction in the amount of C-O-C, C=O groups detected by FTIR, but after thermal treatment it is an increase in the amount of these groups, suggesting oxidation of the composite components.
  5. In Tab. 3, Tg of PLA is 59.8 oC, and ΔHg is 1.716 J/g for blend pellets, whereas, in the text, the data are different, page 11, line298,299.

Please reconsider the result analysis as the differences between the data is small, the decimal numbers are not justified when the temperature 365 oC is considered.

Author Response

Reviewer’s comments

I would recommend the paper “Partial polymer blend for fused filament fabrication with high thermal stability” by Muhammad Harris, Johan Potgieter, Hammad Mohsin, Jim Quan Chen, Sudip Ray, Khalid Mahmood Arif, after minor revision. This paper seems to be a continuation of other works of these Authors.

Please take into account the remarks below:

  1. Would you please check the numbers of tables and figures in the text as they do not fit captions sometimes?

Answer:

Line 236, the mistakenly referred to Table 3 is now deleted.

Line 220, the mistakenly referred to Figure 4 is now corrected to Figure 3.

Line 267, Figure 5 is now cited in the text.

Line 270, the plural of “Figures 5” is now corrected to “Figure 5”.

Line 350: Table 9 is a type error that is now corrected.

  1. The 1% change in transmittance is within the experimental error, so maybe it is not worth comparing such minor changes.

Answer: Lines 283 to 288: The pertinent discussion regarding experimental error is now incorporated into the main text.

  1. Page 9, lines 266-280. The Authors write about a reduction in the amount of C-O-C, C=O groups detected by FTIR, but after thermal treatment it is an increase in the amount of these groups, suggesting oxidation of the composite components.

Answer:

We have noted a writing mistake that has created confusion. The correct changes (calculate) are given as follow,  

Combination

Groups

Non-treated

Thermally treated

Difference

Low temperature combination (161 °C, 25 °C)

C-O-C groups

86%

85%

85%-86%= -1%

C=O groups

90%

85%

85%-90%= -5%

High-temperature combination (171 °C, 85 °C)

C-O-C groups

89.7%

88%

88%-89.7%=1.7%

C=O groups

92%

91.2%

91.2%-92%

Low-temperature treated combination (161 °C, 25 °C)

-1% (85%-86%) in C-O-C groups--- chain scission

-5% (85%-90%) in C=O groups----- probable chemical grafting with MAH

High-temperature combination (171 °C, 85 °C)

-1.7 % (88%-89.7%) of C-O-C --- chain scission

-0.8% (91.2%-92%) of C=O-------probable chemical grafting with MAH

It is noted that both combinations show a decrease in corresponding groups (C-O-C and C=O). The corresponding changes are also added to the main test.

  1. In Tab. 3, Tg of PLA is 59.8 oC, and ΔHg is 1.716 J/g for blend pellets, whereas, in the text, the data are different, page 11, line298,299.

Answer: This was a typo mistake. Now it is corrected. The right glass transition temperature of neat PLA is 65.5, which is now corrected in the Table. Moreover, the ΔHg of non-printed reference blend pellets is now corrected in the text (line 303).

  1. Please reconsider the result analysis as the differences between the data is small, the decimal numbers are not justified when the temperature 365 oC is considered.

Answer: Yes, it is rightly noted. The decimal numbers are not described as main part of the TGA analysis. The minor decimal values are only mentioned in the Table 4 alongside large values to maintain the uniformity in the tables. However, we will surely change as per further suggestions from the reviewer. 

  1. What was the resolution during the recording infrared spectra as the data presented in Figs. 4 and 5 show decimal numbers.

Answer: The decimal values are obtained in FTIR analysis as a result of an average of 30 scans directly provided by the Software (OMNIC E.S.P software version 7.1). In this regard, the excel-based CSV files are used to deduce the corresponding values and to draw the pertinent graphs. 

Reviewer 2 Report

In this study the authors have presented a novel polymer blend of polylactic acid with polypro-pylene for FFF, purposefully designed with minimum feasible chemical grafting and overwhelming physical interlocking to sustain thermal degradation. The content of the work is very rich, mainly based on experiments. The logic of the paper is well organized. The paper can be further modified if it is published.

There are many unsolved questions as below:

  1. The grammar and language should be greatly enhanced. There are too many mistakes. Please check them one by one.
  2. In L36 of P1, “Fused” should be deleted.
  3. In L37 of P1, “renown” should be “renowned”.
  4. In L41 of P1, “pioneer” should be “pioneering”.
  5. In L42 of P1, “utilize” should be “utilizes”.
  6. In L50 of P2, “rational” would not be right.
  7. In L122, there are double “are”.
  8. What is the meaning of the “additional shear”? Is it a type of loading?
  9. There are no equations or formulas in the text. The authors can consider this issue.
  10. The structural stability means “buckling of a slender rod”. Please see: Buckling and wrinkling: Valuable topics in mechanics class. ASCE's Journal of Professional Issues in Engineering Education and Practice, 2018, 144(2): 02518001.
  11. In L190, “KN” should be “kN”.
  12. For the tensile testing, the authors should mention that this is the quasistatic loading. See the related work:  Li et al., Hard to be killed: Load-bearing capacity of the leech Hirudo nipponia. Journal of the Mechanical Behavior of Biomedical Materials, 2018, 86: 345–351.
  13. The tensile strength does not equal to the “stability”.

Author Response

Reviewer’s comments

In this study the authors have presented a novel polymer blend of polylactic acid with polypro-pylene for FFF, purposefully designed with minimum feasible chemical grafting and overwhelming physical interlocking to sustain thermal degradation. The content of the work is very rich, mainly based on experiments. The logic of the paper is well organized. The paper can be further modified if it is published.

There are many unsolved questions as below:

  1. The grammar and language should be greatly enhanced. There are too many mistakes. Please check them one by one.

Answer: Corrected as pointed out.

  1. In L36 of P1, “Fused” should be deleted.

Answer: Corrected as suggested.

  1. In L37 of P1, “renown” should be “renowned”.

Answer: Corrected as suggested.

  1. In L41 of P1, “pioneer” should be “pioneering”.

Answer: Corrected as suggested.

  1. In L42 of P1, “utilize” should be “utilizes”.

Answer: Corrected as suggested.

  1. In L50 of P2, “rational” would not be right.

Answer: Corrected as suggested.

  1. In L122, there are double “are”.

Answer: Corrected as suggested.

  1. What is the meaning of the “additional shear”? Is it a type of loading?

The “additional” is used as “quantitative adjective”. The authors means to explain the adverse effects of shear more than required during extrusion. 

  1. There are no equations or formulas in the text. The authors can consider this issue.

Answer: The formulae are not required here as per best understanding of the content of authors.

  1. The structural stability means “buckling of a slender rod”. Please see: Buckling and wrinkling: Valuable topics in mechanics class. ASCE's Journal of Professional Issues in Engineering Education and Practice, 2018, 144(2): 02518001.

Answer: Structural stability is addresses for one aspect in the referred citation (Buckling). After in depth analysis, the term is found generic, and it is used with different meanings in literature based on type of structural element or component. For example, the referred paper considers the structural stability of Eulerian Rod in terms of buckling. However, the same paper has reported following sentences,

“The most commonly used structures in teaching stability include beams, rods, plates, shells, and arcs, which are elementary building blocks in mechanical, civil, and aerospace engineering structures.”

To explain a few, the structural stability of beams does not provide any buckling. Instead, its structural stability is measured using 3-point or 5-point bending test.

Eisenberger, M., Yankelevsky, D. Z., & Clastornik, J. (1986). Stability of beams on elastic foundation. Computers & structures24(1), 135-139.

The structural stability of plates doesn’t show any buckling. Adding more to the discussion, the term “structural stability” is described using a completely different criteria when studying the polymers. Kindly see the following citation for polymers

Zhang, Y., Li, Q., Welsh, W. J., Moghe, P. V., & Uhrich, K. E. (2016). Micellar and structural stability of nanoscale amphiphilic polymers: implications for anti-atherosclerotic bioactivity. Biomaterials84, 230-240.

Therefore, the structural stability can be translated in different aspects.

In this research, the term “structural stability” is used to present the tensile strength after degradation mechanism (thermal treatment).

  1. In L190, “KN” should be “kN”.

Answer: Corrected as suggested.

  1. For the tensile testing, the authors should mention that this is the quasistatic loading. See the related work:  Li et al., Hard to be killed: Load-bearing capacity of the leech Hirudo nipponia. Journal of the Mechanical Behavior of Biomedical Materials, 2018, 86: 345–351.

Answer: Noted and added in line 190.

  1. The tensile strength does not equal to the “stability”.

Answer: This concern is already addressed in comment # 10.
